# Large-scale brain modes reorganize between infant sleep states and carry prognostic information for preterms

Anton Tokariev [1,2,3], James A. Roberts [1], Andrew Zalesky[4,5], Xuelong Zhao[6], Sampsa Vanhatalo [2,3], Michael Breakspear [1,7,9] & Luca Cocchi [1,8,9]

Sleep architecture carries vital information about brain health across the lifespan. In particular, the ability to express distinct vigilance states is a key physiological marker of neurological wellbeing in the newborn infant although systems-level mechanisms remain elusive. Here, we demonstrate that the transition from quiet to active sleep in newborn infants is marked by a substantial reorganization of large-scale cortical activity and functional brain networks. This reorganization is attenuated in preterm infants and predicts visual performance at two years. We find a striking match between these empirical effects and a computational model of large-scale brain states which uncovers fundamental biophysical mechanisms not evident from inspection of the data. Active sleep is defined by reduced energy in a uniform mode of neural activity and increased energy in two more complex anteroposterior modes. Preterm-born infants show a deficit in this sleep-related reorganization of modal energy that carries novel prognostic information.

[1] QIMR Berghofer Medical Research Institute, Brisbane, QLD 4006, Australia. [2] Department of Clinical Neurophysiology, Clinicum, University of Helsinki, 00014 Helsinki, Finland. [3] BABA center, Pediatric Research Center, Clinical Neurophysiology, Children's Hospital, Helsinki University Central Hospital, 00029 Helsinki, Finland. [4] Melbourne Neuropsychiatry Centre, University of Melbourne, Melbourne, VIC 3053, Australia. [5] Department of Biomedical Engineering, University of Melbourne, Melbourne, VIC 3010, Australia. [6] Department of Bioengineering, School of Engineering & Applied Science, University of Pennsylvania, Philadelphia, PA 19104, USA. [7] Hunter Medical Research Institute, University of Newcastle, Newcastle, NSW 2305, Australia. [8] School of Medicine, University of Queensland, Brisbane, QLD 4006, Australia. [9] These authors contributed equally: Michael Breakspear, Luca Cocchi. Correspondence and requests for materials should be addressed to A.T. (email: anton.tokariev@helsinki.fi) or to L.C. (email: luca.cocchi@qimrberghofer.edu.au)

Changes in behavioral and cognitive states arise from transitions in the configuration of functional brain networks[1–3]. Falling asleep and switching between sleep states are archetypal cortical state transitions that are reflected in the reorganization of patterns of functional interactions between remote brain regions[4,5]. Importantly, the nature of sleep architecture in the early stages of life shapes brain development and influences future behavior[6,7]. Understanding sleep patterns in neonates thus carries important clinical potential[8,9]. However, the neural principles underpinning sleep states early in life are poorly understood. Advances in our knowledge of these fundamental processes are therefore essential for improving brain health outcomes with broader implications for behavior across the lifespan.

Here, we map whole-cortex functional interactions associated with distinct sleep states in two groups of infants, born at term or extremely preterm. We measured cortical activity when infants were in states of active sleep (AS) and quiet sleep (QS). These distinct states of vigilance are key components of the infants' sleep-wake cycle, which gradually transform with neurodevelopment into the alternating cycle of mature rapid eye movement (REM) and non-REM sleep states such as deep sleep[10]. Using high-density electroencephalography (EEG) and tools from network science, we assess frequency-resolved reconfigurations in whole-cortex functional connectivity as a function of sleep states in these two groups of infants. This analysis provides a first comprehensive characterization of abnormal sleep-related reconfigurations of cortical connectivity following extreme preterm birth. To interrogate the notion that infants' sleep states shape the development of cortical pathways and related brain functions, we test if changes in connectivity as a function of sleep states in infants exposed to prematurity carry prognostic information regarding behavioral performance at two years.

Moving beyond the characterization of cortical connectivity patterns, we develop a novel biophysical model based on neural field theory[11]. This is a well-established approach for modeling brain dynamics at macroscopic scales[12] in healthy states[13,14] and following pharmacological manipulation[15]. This model allows study of the neural principles underpinning sleep-dependent changes in cortical dynamics. Our results show that the emergence of distinct sleep-related patterns of cortical connectivity between infants born at term and extremely preterm are caused by differences in the redistribution of energy in low-dimensional modes of spatiotemporal neural activity.

## Results

**Analysis of infant sleep EEG.** We analyzed multi-channel scalp EEG data from two groups of infants: extremely preterm (EP, $N = 42$) and full-term healthy controls (HC, $N = 52$), with gestational age (median ± interquartile range) of $26.6 ± 1.4$ and $40.4 ± 1.7$ weeks, respectively. EEG data in both groups were acquired at term-equivalent age of $41.1 ± 2$ weeks. Continuous epochs of EEG were recorded in states of AS and QS (Fig. 1a). Cortical source signals were reconstructed from these sensor data using a detailed, realistic infant head model[4]. The neuronal signals representing activity of cortical parcels were filtered into four frequency bands: delta (0.4–1.5 Hz), theta (4–8 Hz), alpha (8–13 Hz), and low beta (13–22 Hz). Following standard techniques for the study of functional networks derived from electrophysiological data, we studied a key "intrinsic mode" of connectivity, namely pair-wise amplitude covariation[16]. Frequency-specific functional interactions between cortical activity in these parcels were hence computed as pairwise correlations between amplitude envelopes of mutually orthogonalized neuronal signals[17], resulting in functional connectivity matrices for each frequency band and sleep state (Fig. 1b). We statistically compared[18] whole-brain functional networks between sleep states and groups and correlated these patterns of connectivity with neurocognitive outcomes (Fig. 1c). We then established a biophysical model to explain these cortical network configurations (Fig. 1d).

**Effects of sleep state and preterm birth on brain dynamics.** Analysis of cortico-cortical functional connectivity shows that across both infant groups, sleep states differ significantly in the activity of two broadband networks comprising posterior and anterior brain regions (family wise error rate (FWER)-corrected

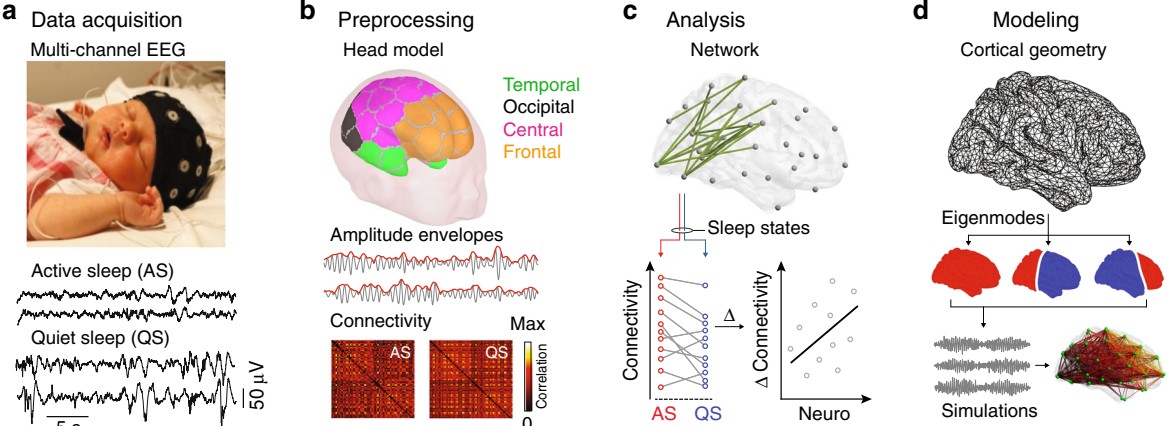

**Fig. 1** Overview of data acquisition and analyses. **a** Multi-channel EEG were recorded at term-equivalent age from 46 infants born extremely preterm (EP) and 67 full-term healthy controls (HC). 5-min-long epochs were selected from two different vigilance states: active sleep (AS) and quiet sleep (QS). EEG during AS is continuous with relative lower amplitude fluctuations, whereas EEG during QS presents as discontinuous (*trace alternant*), with high amplitude signal bursts. Image reproduced with permission from Tokariev et al. (2019). **b** Using an infant head model, cortical source signals were computed from band-pass filtered EEG. The parcellation scheme comprised 58 regions (29 bilaterally symmetric pairs). To assess functional interactions in the brain, we computed correlation coefficients between amplitude envelopes (red lines) of parcel signals (gray lines). This led to connectivity matrices for every infant for both sleep states and for each frequency band (lower panel). **c** Network-based statistics were used to detect patterns of connectivity that statistically differ depending on group or sleep state or both factors. The change of the connectivity strength in the cortical patterns that showed significant interaction was regressed against key neurodevelopmental outcomes of preterm-born infants at 2 years. **d** The geometry of the infant cortical surface (upper panel) was used to compute cortical eigenmodes (lower panel), whose dynamics shape the organization of high-order cortical connectivity

$p_{FWER} < 0.023$ for all paired two-tailed $t$-tests; Fig. 2a for alpha and left column on Supplementary Fig. 1 for other bands). AS is characterized by stronger connectivity within a network comprising occipital, central, and temporal regions (Fig. 2a, b red), extending to frontal cortex in the low beta band (Supplementary Fig. 1). QS shows a more widespread increase in connectivity throughout the cortex, with higher long-range connectivity and a notable involvement of frontal brain regions (Fig. 2a, b blue).

To further characterize the spectral fingerprints of these effects, we filtered the neural signals into (fast) carrier signals and their (slow) amplitude fluctuations across a staircase of relatively narrow bands (evenly spaced in logarithmic coordinates and covering the carrier-amplitude frequency space), then repeated the network-based contrasts. These analyses confirm the existence of functional network effects in both quiet and active sleep states, supported by fast carrier oscillations that are strongest in alpha

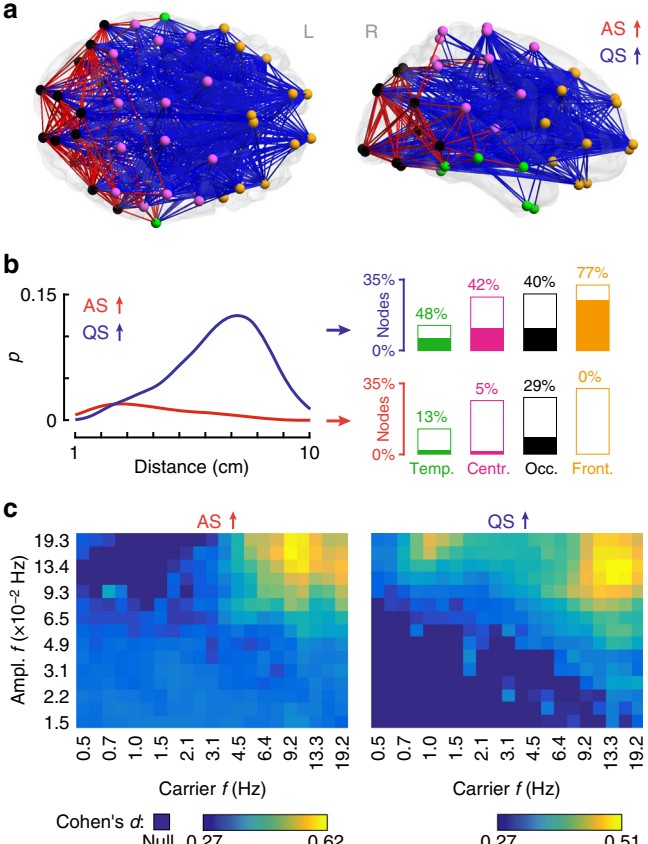

**Fig. 2** Changes in cortico-cortical functional connectivity as a function of sleep state. **a** The main effect of sleep state in the alpha band shows significant connectivity differences in two main cortical networks (red: AS > QS, blue: QS > AS; both $p_{FWER} < 0.0002$, paired two-tailed $t$-test). Distinct cortical regions are coded with different colors: temporal (green), central (pink), occipital (black), and frontal (orange). **b** These networks have different characteristic length distributions and distinct cortical region distributions. Curves are distributions (kernel density estimates) of functional connection lengths given by the fraction of edges $p$ (normalized to the whole network) for AS (red) or QS (blue). Bar plots show participation of broad cortical areas in each sleep-related network. Filling of the bars denotes the percentage of involved nodes within each region and height denotes each region's share in the whole network. **c** Spectral fingerprints of AS (left) and QS (right) sleep. Color scale shows effect size (Cohen's $d$) of sleep contrasts centered at different carrier and amplitude frequencies ($f$). Darkest blue shade denotes frequency combinations where there were no suprathreshold edges (null)

and beta frequencies (Fig. 2c), and whose amplitudes fluctuate maximally on time scales of 5–11 seconds (0.09–0.19 Hz). The strongest effects are seen for higher carrier frequencies in QS than AS (~16 Hz vs. ~10 Hz). Interestingly, there exists a second effect in QS centered over the delta band, composed of relatively short-range edges that involve frontal regions and consistent with the enhanced power of slow-wave activity classically seen in quiet/deep sleep (Supplementary Fig. 1).

Across both sleep states, EP infants show significantly stronger connectivity compared with HC with functional networks in the alpha range encompassing occipital, temporal, central, and frontal regions ($p_{FWER} < 0.0001$, for all paired two-tailed $t$-tests; Fig. 3a, b). These effects are also associated with stronger long-range cortico-cortical connectivity in EP relative to HC (Fig. 3b). Spectral analyses using narrow-band filters suggest this effect is strongest in the alpha and beta range (Fig. 3c), although long-range functional networks, significantly stronger in EP, are also present in the delta and theta bands (Supplementary Fig. 1). EP infants also show a reduction in connectivity between anterior and posterior cortical regions in the theta band ($p_{FWER} = 0.01$, paired two-tailed $t$-test; Supplementary Fig. 1). When analyzed according to the bank of narrow-band filters, the reduction in functional networks is evident across the theta, alpha and beta frequencies (Fig. 3c). However, when analyzed according to the four pre-specified frequency bands, these effects only survive FWER correction in the theta range (Supplementary Fig. 1).

Analysis of interactions between groups and sleep states shows that long-range alpha connectivity between frontal and occipital cortices plays a key role in supporting distinct sleep states that differentiate EP from HC (violet in Fig. 4a, $p_{FWER} = 0.0004$, paired two-tailed $t$-test). Moreover, overlapping patterns of occipital connectivity across a broad frequency range—including theta, alpha, and low beta bands—show a group-by-sleep interaction (orange in Fig. 4a and Supplementary Fig. 2a, all $p_{FWER} < 0.0046$, paired two-tailed $t$-test). For both fronto-occipital and occipital connectivity patterns, the group-by-sleep interaction is driven by an attenuated modulation of connectivity strength between AS and QS in EP (Fig. 4a). Visual inspection of the functional connectivity patterns suggests largely symmetric (left-right) patterns (Figs. 2 and 4a). Formal comparison of the aggregate connectivity within each of the hemispheres confirms that there were indeed no statistically significant left-right asymmetries in either of the groups or sleep states (false discovery rate (FDR)-corrected $p_{FDR} > 0.2$ for all four cases, paired two-tailed Wilcoxon signed-rank test; Supplementary Fig. 3). Comparable results hold true for the low beta band (Supplementary Fig. 2a).

**Sleep-related brain dynamics in ex-preterms preempt outcomes.** We next asked if these group-by-sleep cortical networks carry prognostic information for key developmental outcomes in the EP group. Because of the central role of the occipital cortex in the sleep state differences (Fig. 2a) and the differentiation of HC from EP (Fig. 4a and Supplementary Fig. 2a), we tested whether changes in connectivity across sleep states are linearly associated with visual performance of EP at two years of age (online Methods). To benchmark this against more complex brain functions, we also tested the changes in connectivity against a standard measure of social-emotional performance at two years. These analyses reveal strong and significant (FDR-corrected) negative correlations (partial two-tailed Pearson test controlling for age at EEG recording) between sleep-induced changes in alpha (Fig. 4b; $R = -0.514$, $p = 0.003$, 95% confidence interval (CI) [−0.732, −0.202]) and beta (Supplementary Fig. 2b; $R = -0.461$, $p = 0.009$, 95% CI [−0.698, −0.134]) connectivity in

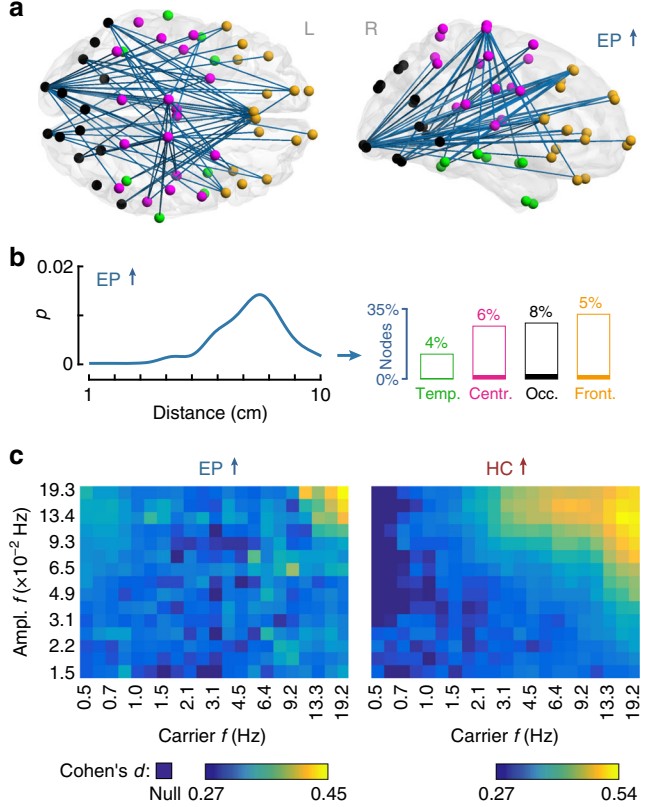

**Fig. 3** Changes in cortico-cortical functional connectivity as a function of birth gestational age. **a** The main effect of group shows higher long-range connectivity in the alpha band in EP infants compared with HC. The corresponding widespread network is composed of temporal (green), central (pink), occipital (black), and frontal (orange) cortices ($p_{FWER} < 0.0001$, paired two-tailed $t$-test). **b** The line shows the distribution (kernel density estimate) of functional connections lengths, where $p$ is the fraction of edges (normalized to the whole network). Bar plots show participation of broad cortical areas in each sleep-related network. **c** Spectral fingerprints of EP (left) and HC (right) infants. Colors show the effect size (Cohen's $d$) of group contrasts for all combinations of carrier vs. amplitude frequencies ($f$). Darkest blue shade marks the cases where there were no suprathreshold edges (null)

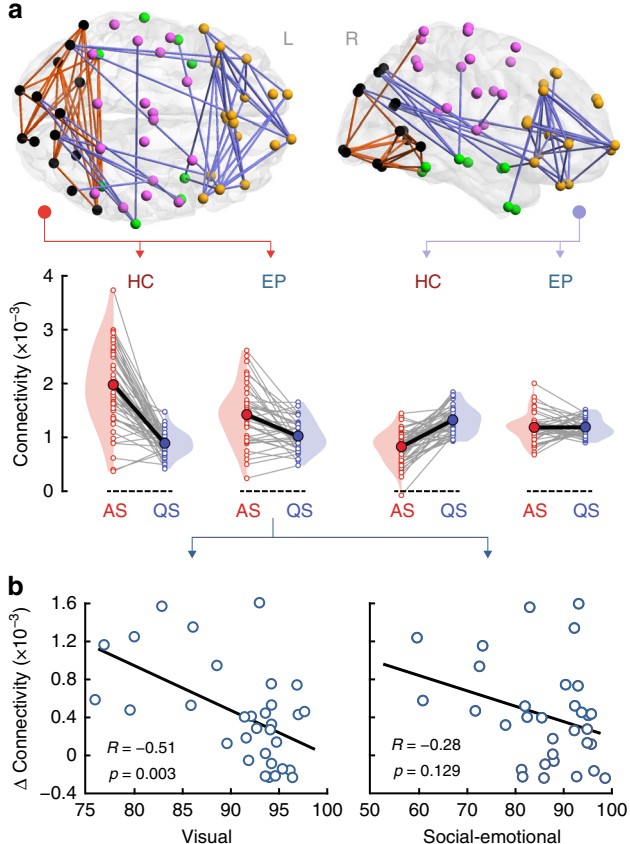

**Fig. 4** Alpha-band cortical networks showing a sleep-by-group interaction and their association with neurocognitive outcome at two years. **a** A positive interaction (changes in connectivity between AS and QS in HC vs. EP) is found for patterns of connectivity linking occipital regions, with only a few connections to temporal and central cortices (orange; $p_{FWER} = 0.003$, paired two-tailed $t$-test). This pattern is spatially consistent also across neighboring theta and low beta frequency bands (Supplementary Fig. 2a). A negative interaction (opposite changes in connectivity between AS and QS as a function of group) is observed for long-range connectivity between the frontal cortex and occipital regions (violet; $p_{FWER} = 0.0004$, paired two-tailed $t$-test). These interactions reflect attenuated changes in connectivity between sleep states in EP compared to HC. Gray lines link individual infants and thick black lines indicate slopes between group means. **b** Correlations between changes in the mean cortical connectivity strength of the occipital pattern (connectivity strength of the orange network of panel **a** in AS minus QS) and neurodevelopmental outcome assessed at two years of age in EP infants. Changes in alpha connectivity correlate significantly with visual performance ($R = -0.514$, $p = 0.003$, surviving FDR, partial two-tailed Pearson's correlation). The association with social-emotional outcome is not statistically significant. Source data are provided as a Source Data file

posterior networks (comprising mostly occipital cortices) with visual performance. No significant associations between sleep-induced connectivity changes and visual behavior are detected for low-frequency bands. A negative correlation is also apparent for social-emotional performance, but the association is not statistically significant when accounting for multiple comparisons (all $p_{uncorrected} > 0.039$; Fig. 4b and Supplementary Fig. 2b). The regressions against socio-emotional indices and functional networks are not significant in any frequency band, nor are those for visual outcome and functional networks in the theta band (Supplementary Fig. 2b). These findings support the conjecture that preterm birth has a major effect on the development of large-scale neural networks underpinning behavior[6], including later neurocognitive performance that relies on visual function[19].

**Cortical eigenmodes shape early brain activity.** It has been recently shown that large-scale patterns of neural activity can be modeled with cortical eigenmodes[11,13,20]. That is, that the geometry of the cortex constrains the decomposition of neural activity into spatiotemporal modes—analogous to the harmonics of a musical instrument—whose linear superposition forms the basis of whole-brain functional connectivity patterns. The

dynamic activity that each spatial mode carries can be modeled through the application of neural field theory[14,21]. The functional networks that differ between AS and QS have the appearance of a long-wavelength anteroposterior pattern of activity that changes between the two sleep states (Fig. 2a). To assess if this broad anteroposterior pattern of cortical connectivity emerges from changes in the activity of key low-dimensional modes, we applied an eigenmode decomposition to a neonatal cortical mesh (see Methods). Similar to the eigenmodes calculated on the adult cortex[11], these modes possess either a symmetric or antisymmetric appearance, explaining increasingly complex patterns of activity as the mode order increases (Fig. 5a and Supplementary

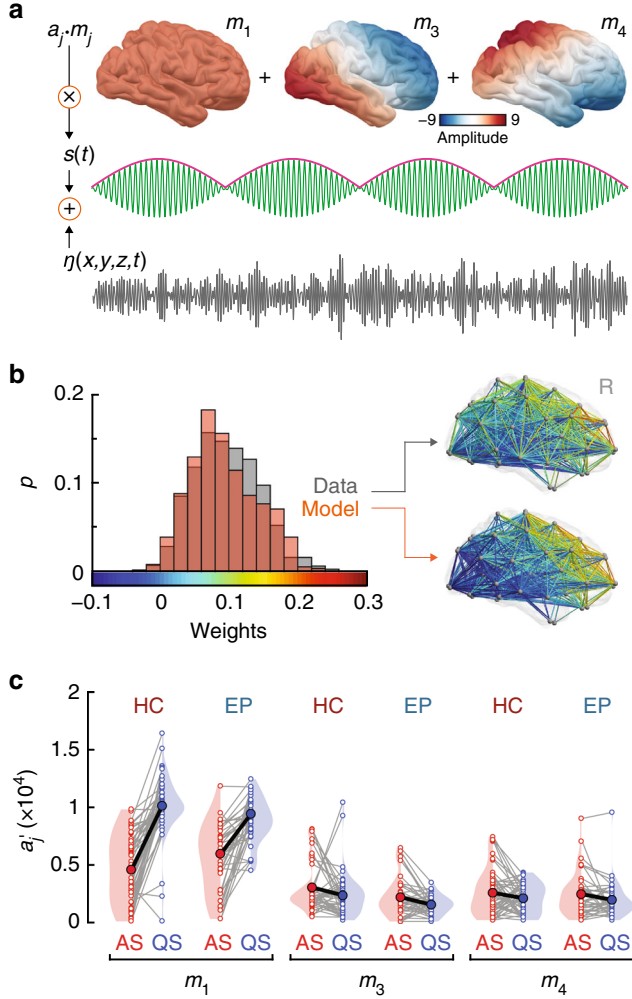

**Fig. 5** Analysis of low-dimensional modes of spatiotemporal neural activity as a function of sleep states. **a** Schematic representation of the neural field model adopted to investigate the principles underpinning patterns of functional connectivity associated with neonatal sleep states. The model is composed of cortical eigenmodes[11], a low-frequency signal amplitude modulation (pink), a faster carrier frequency (green), and spatiotemporal Gaussian white noise. The first six modes are depicted in Supplementary Fig. 4. **b** Comparison between empirical data (control infant in AS) and fitted model functional connectivity. The histograms show the probability distributions of connectivity weights of all network edges in the data (gray) and model (orange). The best fit minimizes the sum of the squared differences between the modeled and empirical distribution means and SDs (Methods). The brains depict the spatial distributions of the connectivity weights for the data and model fit. Supplementary Fig. 5 provides a broader overview of the fitting between model and real data across representative subjects. **c** Mode ($m_j$) weights ($a'_j$) calculated across the three modes as a function of sleep states and group. Gray lines link individual subject weights (open circles), thick black lines link state-specific group mean weights (filled circles). Significant main effects of sleep are observed for the three modes of interest (mixed ANOVA: mode 1: $p = 8.2 \times 10^{-22}$, mode 3: $p = 0.006$, mode 4: $p = 0.021$). A sleep-by-group interaction is present in mode 1 ($p = 0.006$). Source data are provided as a Source Data file

Fig. 4). The first mode ($m_1$) is uniform and global, whereas the second mode ($m_2$) is a left-right pattern reflecting the symmetry of the brain's hemispheres. The next two modes ($m_3$ and $m_4$) capture long-range anteroposterior activity patterns, which mirror the observed changes in cortical activity as a function of sleep state (Fig. 2a).

The dominant anteroposterior orientation and lack of fine spatial structure of the connectivity contrasts depicted in Figs. 2 and 4 suggests an explanation in terms of the relative weights of these four low order modes. This is plausible because low order modes are the most excitable, and hence carry the most energy[22]. To test the hypothesis that changes in sleep-related connectivity reflect changes in the weights of the eigenmodes, we developed a parsimonious neural field model of cortical activity in which simulated neural activity, $Y$, is a linear superposition of spatial eigenmodes, each of which supports fast carrier oscillations that fluctuate in amplitude on slower time scales. The functional connectivity of these oscillations across the cortex emerges from the spatial character of each corresponding mode. To achieve this, we express $Y$ as,

$$Y = \sum_j a_j m_j(x, y, z) \cos(\nu t) \cos(\omega t) + \sigma \eta(x, y, z, t), \quad (1)$$

where $Y = Y(x, y, z, t)$ is the modeled spatiotemporal cortical activity at time $t$ and position $(x, y, z)$, $a_j$ are the mode weights, $m_j$ are the mode spatial patterns, $\nu$ is the (low) frequency of amplitude modulation, $\omega$ is the faster carrier frequency, $\eta$ is spatiotemporal Gaussian white noise, and $\sigma$ is its standard deviation (Fig. 5a). To model the effects in the alpha band, we set $\omega$ to 10 Hz, with amplitude modulated on the slow time scale of ~10 s ($\nu = 0.1$ Hz). To match the sampling of the empirical data, cortical dynamics were simulated for 300 s at 100 Hz sampling frequency. We then determined the subject-specific values of $\sigma$ and the $a_j$ mode weights such that the modeled activity best matches the empirical data. The best fitting parameters minimize differences in the first two moments of the empirical vs. the modeled distributions of functional connectivity values. Connectivity only depends on the relative sizes of the oscillations and the noise; the overall scaling of the model dynamics is set to match the standard deviation (SD) of the empirical time series. As there are no left-right asymmetries in our functional networks (Supplementary Fig. 3), we further simplified the model by setting the second mode to zero. We therefore focus on bilaterally symmetric modes, estimating the coefficients for the first ($m_1$), third ($m_3$), fourth ($m_4$) modes (Fig. 5a). The face validity of our model is highlighted by a strong overlap between individual empirical and simulated patterns of cortico-cortical connectivity (Fig. 5b and Supplementary Fig. 5). These results indicate that the observed functional connectivity patterns across sleep states (Fig. 2) arise from the distribution of energy in oscillatory spatiotemporal modes and their ratio to unstructured noise.

**Sleep states reflect redistribution of energy in brain modes.** Fitting the mode parameters to our empirical data reveals that sleep state transitions predominantly reflect a change in the weighting of the first mode with significant but weaker changes in the coefficients of the third and fourth modes (Fig. 5c). Compared to active sleep, quiet sleep is defined by higher energy in the first uniform mode ($F_{1,92} = 1.6 \times 10^2$, $p = 8.2 \times 10^{-22}$, effect size $\eta^2 = 0.615$, mixed ANOVA) and slightly lower energy in the two anteroposterior modes (mode three $F_{1,92} = 7.98$, $p = 0.006$, $\eta^2 = 0.08$; mode four $F_{1,92} = 5.51$, $p = 0.021$, $\eta^2 = 0.057$; mixed ANOVA; Fig. 5c). Hence, the model indicates a shift in the weighting of uniform to more complex antero-posterior activity across sleep states, and a re-weighting of these broadly coherent oscillations relative to the noise.

We found a significant sleep-by-group interaction confined purely to the first mode (mixed ANOVA, $F_{1,92} = 7.76$, $p = 0.006$, $\eta^2 = 0.03$; Fig. 5c). This result suggests that dynamics in the first, uniform mode underpins the different patterns of high order-connectivity observed in HC vs. EP as a function of sleep states.

That is, a decrease of energy in this mode reveals relatively greater energy in the higher order modes—and more so in HC than in EP—and hence the correspondingly more complex spatiotemporal patterns evident in Fig. 2. Importantly, this reduction in global cortical energy in EP is also reflected in spectral power changes across sleep states in empirical data (Supplementary Fig. 6).

## Discussion

We combined connectivity analyses of high-density EEG with biophysical modeling to study cortical dynamics associated with sleep states in pre-term and full-term infants. We found that sleep-related differences in connectivity patterns between infants born at term vs. those born extremely preterm correlate with later neurodevelopmental outcomes. Crucially, biophysical modeling pinpointed a specific deficit in large-scale neuronal dynamics underpinning these sleep-induced connectivity differences. Understanding the neural basis of infant sleep is an important endeavor for basic and translational neuroscience. Our work establishes several new systems-level insights in this endeavor, and holds potential for clinical translation.

We found that infant AS vs. QS states are characterized by broadband differences in cortical connectivity. AS is associated with increased functional connectivity in visual occipital regions whereas QS involves a broader increase in long-range cortical connectivity. These modulations of whole-cortex connectivity across different sleep states support the proposal that sleep-dependent patterns of brain activity play a key role in promoting the development of large-scale cortical networks that underlie a broad range of behaviors[4,23]. As a specific example of this, we find that changes in occipital connectivity between AS and QS are affected by extremely preterm birth, and the magnitude of this alteration correlates with visual behavioral outcomes at two years of age. That is, alterations in functional connectivity in visual cortex due to preterm birth may impact on the maturation of functional assemblies that are important for emerging visual function. Although we find enhanced correlations in lower frequencies during QS, as might be expected for activity during deep sleep[24], these do not correlate significantly with neurobehavioral outcomes. Rather, our findings support the notion that synchronization within the visual system at alpha and beta frequencies is key for the development of perceptual function[25,26]. The correlation of occipital functional connectivity with social-emotional development at two years is not statistically significant, suggesting some specificity to these functional-developmental couplings. Intriguingly, the correlation between sleep-state related functional network changes in the EP infants and visual function at 2 years (Fig. 4b) is in the opposite direction to the between-group effect with the HC infants. Without experimental control over the exposure of EP infants to normal sensory experience in the neonatal intensive care unit, it is not possible to infer whether these effects are driven by preterm birth or the additional postnatal exposures of EP infants. Future work, incorporating longer-term (life-long) outcomes and other functional metrics could also help disambiguate those preterm-related functional network deficits that persist from those that can be modulated by environmental influences during childhood (e.g., inter-personal relationships, health interventions, education).

Computational modeling suggests that a redistribution of energy in low-dimensional modes of spatiotemporal neural activity supports the emergence of these sleep-related patterns of cortical functional connectivity. Specifically, the broad increase in cortical connectivity associated with QS emerges from an increase of oscillatory energy in a globally uniform mode. The regional increase in occipital connectivity in AS reflects a relatively greater proportion of energy in higher-order oscillatory modes following a reduction in the first mode. Of note, changes in the corresponding more complex antero-posterior modes—whose geometry mirrors the topology of the functional connectivity changes—are small in comparison. The use of computational modeling thus highlights changes in large-scale cortical activity that are not immediately obvious from inspection of functional connectivity changes viewed in isolation, but instead speak to more subtle changes in whole brain oscillatory and stochastic activity.

Scalp EEG is most sensitive to source activity in adjacent cortex. While sleep-related cortical reconfigurations are likely driven by subcortical structures[27], our findings permit a non-invasive characterization of the cortical manifestations of these subcortical modulations. Moreover, these results relate whole-cortex dynamics to functional brain development. The physiological mechanism for the transitions themselves—i.e., the mechanisms for the dynamics of the mode weights—could in principle be described using models of the ascending arousal system[28,29]. These ascending neuromodulatory systems modulate neuronal gain, increasing neuronal signal to noise in a manner that supports abrupt phase transitions[30]. The present findings thus provide a foundation for the development of a unified theory of neural dynamics supporting sleep. The integration of invasive and non-invasive neurophysiological recordings in preclinical models of preterm birth could play an important role in overcoming the inevitable limitation of scalp-only EEG in human clinical data. Likewise, our eigenmode decomposition is currently derived from cortical geometry only: The incorporation of structural connectomes of infant brain into the decomposition[13] could further refine and individualize the functional modes used for our biophysical model. Such work requires further advances in the accurate reconstruction of individual tractography data from neonates. Given that geometry contributes substantially to connectivity[31], which shapes large-scale dynamics[32], it is likely that such resulting modes would differ only slightly from those presently used.

Findings from the current work have substantial translational potential, opening a novel diagnostic and prognostic window into brain monitoring and prediction of developmental outcomes in extremely preterm infants[33]. This information could be harnessed to develop new bed-side tests needed for benchmarking interventions to improve the neurocognitive outcomes of preterm infants, the globally biggest neonatal risk factor[34]. More broadly, our framework shows how the reconfiguration of functional networks across brain states can be recast as changes in large-scale spatiotemporal modes of activity. Together with the rapid penetration of brain network theory across clinical disorders[35], this suggests a fundamental role for neural field theory in translational neuroscience.

## Methods

**Ethics**. The study design and procedures have been approved by the Ethics Committee of the Helsinki University Central Hospital (Finland). Informed written consent was received from a guardian before inclusion of an infant into the study.

**Subjects**. Data were acquired at the Helsinki University Central Hospital (Finland). Some data related to infants included in this cohort have been used for previous independent publications[4,7,36,37]. Multi-channel EEG data were collected from two cohorts of infants: extremely preterm (EP, $N = 46$) and full-term healthy controls (HC, $N = 67$).

**EEG data**. EEG data were recorded with a NicOne EEG amplifier (Cardinal Healthcare/Natus, USA) or a Cognitrace amplifier (ANT B.V., Enschede, The Netherlands). EEG caps (sintered Ag/AgCl electrodes; Waveguard, ANT-Neuro, Germany) had 19 or 28 scalp electrodes positioned according to the International 10–20 standard. The same 19 EEG channels (Fp1, Fp2, F7, F3, Fz, F4, F8, T7, C3, Cz, C4, T8, P7, P3, Pz, P4, P8, O1, O2) across all recordings were selected for analyses. Further details regarding EEG acquisition in newborns can be found elsewhere[38].

**Sleep state selection**. Each recording session continued until the infant had undergone two sleep (vigilance) states: active sleep (AS) and quiet sleep (QS). Vigilance state assessment was performed in our data through a combination of electrophysiological and behavioral measures. Polygraphic channels (chin electromyogram, electrocardiogram, electrooculogram, and respiratory sensors) were used for this purpose. EEG traces during AS exhibit continuous fluctuations, respiration is irregular, and occasional eye movements are present. Conversely, EEG during QS is characteristically discontinuous, and respiration is regular[39] (Fig. 1a). Next, we selected 5-min-long artifact-free EEG epochs from the most representative periods of AS and QS. The selection of the epoch length was based on our previous studies demonstrating that a continuous 5 minute recording duration provides a reliable and stable estimate of the whole-brain functional network activity in infants[4,7]. To avoid transitions between vigilance states, representative sleep epochs were selected from within well-established patterns of corresponding behavior and brain activity, and not close to the transition between QS and AS. Subjects that lacked sufficient epoch lengths or had poor quality data (due to movement artifacts or loss of contact) were excluded from further analysis. The final sample included 42 neonates in the EP group and 52 neonates in the HC.

**Neurodevelopmental assessment**. The neurodevelopment of infants was assessed at two years of corrected age using the structured Griffiths Mental Developmental Scales[40]. These scales were chosen because they measure the cognitively demanding domains of visual performance and social-emotional performance. The assessment of visual and social-emotional performance in EP was chosen because of their established, widespread clinical use plus their broad impact on lifelong neurocognitive performance and quality of life, hence improving the translational potential of our study[41,42]. While other outcomes such as gross motor development and hearing are also affected in some infants, studies in ex-preterm infants have shown that these are the most likely to be modified by a host of individual and treatment interventions[43].

**EEG data pre-processing**. EEG data were initially band-pass filtered into the 0.15–45 Hz frequency band, down-sampled to a sampling rate of 100 Hz and re-referenced to the common average reference montage. For further analysis we filtered the data into four frequency bands of interest: 0.4–1.5 Hz (delta, $\delta$), 4–8 Hz (theta, $\theta$), 8–13 Hz (alpha, $\alpha$), and 13–22 Hz (low beta, $\beta$). For band-pass filtering, we applied in series a combination of low-pass and high-pass Butterworth filters with the corresponding cut-off frequencies and stop-band attenuation of 20 dB. Each filter was applied in both forward and backward directions to avoid introducing phase lags into the EEG signals. An overview of the entire analytic pipeline is shown in Supplementary Fig. 9.

**Source reconstruction**. A realistic infant head model was used to compute cortical source signals from multi-channel EEG[4]. Briefly, we generated scalp, skull, and intracranial volume shells (2562 equidistant vertices in each) from segmented anatomical magnetic resonance imaging data of a healthy full-term infant. In the source space, we used a cortical template scaled to the infant size and spatially smoothed to match the brain folding at term-equivalent age. The source space comprised 8014 electrical dipoles (of fixed orientation orthogonal to the surface) approximating the local neuronal activity. Tissue conductivities for intracranial volume, skull, and scalp were taken as 1.79 S/m, 0.2 S/m, and 0.43 S/m, respectively[44,45]. The head model included 19 scalp EEG electrodes placed according to the empirical recordings. The forward solution (i.e., the operator that estimates the contribution of cortical sources to scalp EEG) was computed using a symmetric boundary element method implemented in the openMEEG package[46]. For the noise covariance matrix, we used the identity matrix, which assumes equal noise levels in EEG sensors. To compute cortical sources from EEG (the inverse solution), we applied dynamic statistical parametric mapping[47] as implemented in the Brainstorm software package[48]. The 8014 cortical sources were clustered into 58 parcels including $125 \pm 26$ (mean $\pm$ SD) sources each (Fig. 1b). The adopted brain parcellation scheme was symmetric across hemispheres (29 parcels in each hemisphere). The activity of each parcel was taken as the weighted mean activity of all sources within it[4].

**Connectivity analysis**. Pairwise functional interactions between cortical parcels were estimated by computing Pearson correlation coefficients between amplitude envelopes of the corresponding cortical signals. Pairs of signals were first orthogonalized relative to each other in 2 s non-overlapping time windows[17]. This procedure was performed in both directions: signal $X$ was orthogonalized relative to signal $Y$, and signal $Y$ was orthogonalized relative to $X$. Two output correlation coefficients were then averaged and used as the functional interaction estimate between $X$ and $Y$. The correlation coefficients between all possible parcel pairs led to a full adjacency matrix having $(58 \times 57)/2 = 1653$ connections. The resulting matrices were corrected by excluding connections which cannot be reliably estimated using 19 recording EEG electrodes. To define such connections, we simulated artificial parcel activity, where pairs of specific regions were synchronized at one time, and that was used to compute synthetic EEG and to reconstruct back parcel signals. We repeated this in 500 iterations for each pair of parcels. Finally, we contrasted pairwise interactions of reconstructed parcels that were initially in

synchrony to the set of surrogate values from all non-synchronous parcels from all iterations. This allowed us to generate a statistically based binary template where all interactions below the 99th percentile of surrogates were rejected (zeros) as non-reliable and all others (ones) were used for further analysis (for more details see elsewhere[4]). Here, we excluded 32% of interactions from all adjacency matrices (the same non-reliable interactions in all subjects), leaving 1128 connections in each for further analysis. Finally, to reduce inter-individual differences in total connectivity across subjects, each adjacency matrix was normalized by dividing each weight by the global sum of the magnitude of the weights in the matrix.

**Network-based statistics**. The network-based statistic (NBS)[18] was used to estimate group and sleep main effects, as well as group-by-sleep interactions. Analyses were carried out independently for each frequency band of interest. A search $t$-statistic threshold of 3 (equivalent to $p < 0.001$, uncorrected) was initially applied to all 1128 possible pairwise connections between cortical parcels. The size of surviving sets of pairwise connections was recorded. Permutation testing was used to estimate a corrected $p$-value for each pattern of connections ($p < 0.05$, FWER-corrected at the level of the whole pattern). In each permutation, the assignment of data to a given group or sleep state was randomized. For each of the 5000 permutations the size of the largest network was recorded, allowing for the generation of a null distribution and FWER correction.

**Assessment of effect size in the frequency domain**. To study the spectral fingerprints of sleep and gestational age effects, pre-processed EEG data were filtered into 21 frequency bands covering the range 0.42–22 Hz. Central frequencies ($f$) of these filters were spaced such that the $(k + 1)$-th frequency was a fixed multiple of the $k$-th frequency given by $f_{k+1} = 1.2 f_k$. The slowest carrier frequency $f_1$ was set to 0.5 Hz and the ensuing highest frequency was $f_{21} = 19.2$ Hz. Cut-off frequencies were $0.85f$ and $1.15f$, and stop-band frequencies were $0.5f$ and $1.5f$. Band-pass filtering was implemented by applying pairs of low-pass and high-pass Butterworth filters. This yielded carrier oscillations with overlapping frequency bands of equal width on a logarithmic scale. The amplitudes of these carrier frequencies were then band-pass filtered into 15 frequency bands to estimate amplitude fluctuations at different time scales[49]. Amplitude filters were designed according to the same principles as for the carrier frequencies. The slowest amplitude frequency $f_1$ was set to 0.015 Hz and the ensuing highest frequency was $f_{15} = 0.19$ Hz. This double filter bank led to $21 \times 15 = 315$ functional connectivity matrices per subject at each sleep state. Connectivity analyses of the main effects of group and sleep state were estimated using NBS with a search (height) $t$-statistic threshold of 2.5. Frequency combinations (carrier and amplitude) for which there were no suprathreshold edges were denoted null (Figs. 2c, 3c). Effect size was computed as Cohen's $d$ (i.e., NBS-derived mean $t$-statistic divided by the square root of the total degrees of freedom).

**Brain-behavior correlations**. For each sleep state (AS and QS) and EP infants with follow-up neurocognitive examination ($N = 32$), we computed the mean connectivity strength in cortical patterns showing a significant group-by-sleep interaction (Fig. 4a and Supplementary Fig. 2a). We then calculated subject-specific differences in mean connectivity (AS minus QS). Finally, we computed correlation coefficients (two-tailed Pearson's correlation) indexing the linear association between changes in functional connectivity and scores of visual and social-emotional performance at two years of age. We used partial correlation analyses (*partialcorr* function in Matlab), controlling for conceptional age of infants at the time of EEG recording to account for neurodevelopment[7]. Resulting p-values from the six brain-behavior tests (two behavioral measures and three frequency bands) were corrected using a standard Benjamini-Hochberg false discovery rate (FDR) correction procedure.

**Eigenmode decomposition and computational model**. Application of neural field theory has recently shown that large-scale resting-state activity patterns are well described by eigenmodes of the cortical surface[11]. Details regarding neural field theory[14,22] and solutions of neural field equations on cortical surfaces to obtain eigenmodes[20] have been extensively presented elsewhere[11]. Here, we briefly describe the specific aspects of neural field theory employed, and the practical steps to calculate eigenmodes.

Neural field theory is a continuum description of neural activity, built upon standard (mean-field) approaches from physics and used to average over the microscopic details of individual neurons[50]. This enables modeling of the dynamics of large populations of neurons, where the relevant physiological quantities are average membrane potentials and firing rates over local neural populations. For comparison with EEG, the main quantity of interest is the excitatory cortico-cortical activity field $\phi_{ee}$, representing the mean incoming spike rates at pyramidal neurons. We are interested in spatiotemporal dynamics and hence use a formulation in terms of partial differential equations, which retain spatial information across the cortical sheet modeled here as a two-dimensional surface. Importantly, the general form of the model adopted here can be used to accurately predict resting-state EEG spectra[14,51], including sleep[52]. It has been shown that, in the absence of external inputs, resting-state cortical activity can be written as a sum

of eigenmodes

$$\phi_{ee} = \sum_j m_j(\boldsymbol{r})e^{-i\omega_j t},\qquad(2)$$

each with eigenfrequency $\omega_j$, and amplitude $m_j(\boldsymbol{r})$ at position $\boldsymbol{r}$ satisfying the Helmholtz equation,

$$\nabla^2 m_j(\boldsymbol{r}) = -k_j^2 m_j(\boldsymbol{r}),\qquad(3)$$

where $k_j^2$ is the eigenvalue and $\nabla^2$ is the Laplace-Beltrami operator on the cortical surface. We used a cortical surface[53] scaled to the infant size and spatially smoothed to match the brain folding at term-equivalent age. We solve Equation (3) for the eigenmodes using a finite element method implemented in Matlab[54–56]. The first six eigenmodes are shown in Supplementary Fig. 4.

The existence of amplitude-amplitude correlations in the data requires the oscillatory amplitudes to vary in time at each point. We assume the simplest form of oscillatory amplitude modulation in Equation (1) of the main text, where each mode is modulated with a (cosine) oscillation at a fixed low frequency $\nu$ (here, 0.1 Hz). This is equivalent to a beat, a linear superposition of two nearby frequencies. Thus, each amplitude-modulated mode $m_j \cos(\nu) \cos(\omega t)$ in our final signal $Y$ is a sum of a pair of pure oscillatory modes $\frac{1}{2} m_j \cos\left[(\omega - \nu)t\right] + \frac{1}{2} m_j \cos\left[(\omega + \nu)t\right]$, where we have also assumed that each amplitude-modulated mode has the same carrier frequency $\omega$ (here, 10 Hz for alpha band oscillations).

We restricted our attention to the first four modes $j = 1$–$4$ (Supplementary Fig. 4) on the basis that the dominant feature in Fig. 2a is a clear anteroposterior pattern, with an absence of the finer spatial structure captured by modes $j \geq 5$. Moreover, as detailed in the main text, formal testing for a left-right asymmetry in the functional connectivity patterns showed no significant effect (Supplementary Fig. 3). Hence, we further simplified the model by setting mode weight $a_2 = 0$. This leaves four parameters in Equation (1) that need to be estimated from the data: $a_1$, $a_3$, $a_4$, and $\sigma$.

We fitted the model using a two-step procedure. First, because the correlation dynamics only depend on the ratios of the $a_j$ to the noise amplitude $\sigma$, without loss of generality we fixed $\sigma = 1$ and calculated the model functional connectivity over a 3-D grid of values of $a_1'$, $a_3'$, and $a_4'$, where the primes denote the $\sigma = 1$ case. The gridded values of $a_j'$ spanned the range 0-0.5 (steps of 0.0025). This range was selected based on initial testing that showed that the best fits fall in the range $0 < a_j' < 0.5$ (Supplementary Fig. 7). We summarized each pattern of functional connectivity in the parameter space by calculating the first two moments (mean and SD) of the distribution of edge weights. Then, for each subject, we calculated the same summary statistics and found the corresponding parameters that minimized the cost function,

$$J(a_1', a_3', a_4') = [\mu_{\mathrm{data}} - \mu_{\mathrm{model}}(a_1', a_3', a_4')]^2 + [\sigma_{\mathrm{data}} - \sigma_{\mathrm{model}}(a_1', a_3', a_4')]^2,\qquad(4)$$

where $\mu_{\mathrm{data}}$ and $\sigma_{\mathrm{data}}$ are the mean and standard deviation of the functional connectivity weights across the empirical network, and $\mu_{\mathrm{model}}$ and $\sigma_{\mathrm{model}}$ for the model. We finally estimated the subject-specific $\sigma$ and $a_j = \sigma a_j'$ by finding the value of $\sigma$ such that $Y(t)$ and the empirical time series have the same standard deviation across time.

Data from all ($N = 42$) preterm infants were used in these analyses. For display purposes, data from one outlier was omitted from Fig. 5c (see Supplementary Fig. 8 for all data points). This data point was included in all formal analyses.

**Statistical analysis**. Pairwise group comparisons for connectivity asymmetries (Supplementary Fig. 3) were performed using two-tailed signed-rank Wilcoxon tests. This non-parametric approach was adopted because data did not follow a normal distribution (Shapiro-Wilk test). $P$-values were FDR corrected with Benjamini-Hochberg procedure.

To test for significant differences in mode weights (Fig. 5c) and global amplitudes (Supplementary Fig. 6), we used a mixed analysis of variance (ANOVA).

**Analysis software**. MRI segmentation was performed using FSL[57] and the head model was computed using the openMEEG package[46]. Source reconstruction was done using algorithms implemented in Brainstorm[48]. Brain networks were visualized using the Matlab toolbox BrainNet Viewer[58]. Statistical analyses were performed using standard and custom Matlab functions and JASP software (https://jasp-stats.org).

**Reporting summary**. Further information on research design is available in the Nature Research Reporting Summary linked to this article.

## Data availability

Data will be available upon reasonable request. The source data underlying Figs. 4, 5 and Supplementary Figs. 2, 3, and 6 are provided as a Source Data file.

## Code availability

Custom Matlab code implementing connectivity analysis, the biophysical model, and its fitting to empirical data can be found here: https://github.com/babyEEG/Infant-Sleep.

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

## Acknowledgements

The authors thank Dr. Aulikki Lano and Dr. Mari Videman for assessing neurodevelopment of the infants, and A.J.K. Phillips for helpful comments. A.T. was supported by Finnish Cultural Foundation (Suomen Kulttuurirahasto; 00161034). A.T. and S.V. were also funded by Academy of Finland (276523 and 288220) and Sigrid Juséliuksen Säätiö (Sigrid Juséliuksen Säätiö), as well as Finnish Pediatric Foundation (Lastentautien tutkimussäätiö). J.A.R., A.Z., M.B. and L.C. are supported by the Australian National Health and Medical Research Council (J.A.R. 1144936 and 1145168, A.Z. 1136649, M.B. 1037196, L.C. 1099082 and 1138711). This work was also supported by the Rebecca L. Cooper Foundation (J.A.R, PG2018109) and the Australian Research Council Centre of Excellence for Integrative Brain Function (M.B., CE140100007).

## Author contributions

A.T., J.A.R., S.V., M.B., L.C. designed the research; A.T., J.A.R., L.C. performed the analyses; A.Z. and X.Z. contributed analysis tools; S.V. organized data collection; and all authors wrote the paper.

## Additional information

**Competing interests:** The authors declare no competing interests.

