## [Peer Review File · Nature Communications]

Reviewers' comments:

Reviewer #1 (Remarks to the Author):

In this paper, the authors investigate cortical dynamics associated with sleep states in pre- and full-term infants by using a very clever connectivity analysis of EEG and more importantly a model based study of the underlying eigenmodes. The paper is, in my view, extremely relevant not only for basic neuroscience but also for clinical applications aiming to predict the development of pre-term neonates. More importantly, I think they show very specifically the necessity of using whole-brain modelling for extracting the biophysical underlying causative mechanisms of the observed changes.

In particular, they show significant changes in the global brain connectivity underlying the babies AS and QS sleep stages differentiating the control normal babies respect to the pre-term neonates. Further more, they were able to demonstrate that those significant changes in global connectivity underlying different sleep stages are due to a specific deficit in large-scale neuronal dynamics.

More specifically, they observed that AS sleep is associated with increased functional connectivity in visual occipital regions whereas QS is associated with a broader increase in long-range cortical connectivity.

The paper is very well written and the methodology proposed is novel and solid, so I'm absolutely convinced that this paper should be accepted in Nature Communications due to its novelty, relevance and elegant combination of empirical phenomenological analysis with causative model-based explanations.

I have only some minor remarks that the authors could decide to consider:

1) I think that would be nice to analyse the data also in narrow bands (let say in windows of 2Hz in order to have a more specific frequency analysis. Even more, perhaps after the first narrow band filters , it would be nice to analyse the functional correlation of the envelopes after a second slow filte (let say from 0.01 Hz to 0.2 Hz). the motivation is the type of analysis that one use in MEG (e.g. Hipp et al. Nature Neuroscience volume 15, pages 884–890 (2012)). Perhaps one can also discover the relevant time scale (associated with this second filter), i.e. for which time scale the method is more sensitive.

2) I assume that based on the eigenmodes of the anatomical connectivity matrix, one could use those templates for also analyzing the correlation patterns and their changes on those modes. I mean to use the modes described in Selen et al. Nature Communications volume 7, Article number: 10340 (2016). Of course , I'm NOT asking to do these analyses, that would be a new paper, but perhaps to discuss/speculate about them in the conclusions as alternative, or to argue why functional modes are more suitable than tose anatomical eigenmodes.

Excellent paper, I enjoyed reading that.

Reviewer #2 (Remarks to the Author):

The authors analyzed brain cortical connectivity between infant sleep states (active sleep/quiet sleep) and between extremely premature neonates and term infants. They also statistically analyze the sleep state-related connectivity as prognostic information of visual performance and social-emotional performance later at around 2-year age. The work was very interesting with several significant findings and the paper is well-written. Statistical analyses were appropriate.

- It can be better explained or motivated the choice of the visual and social emotional performance at age of 2 y for analysis. There are other aspects of the mental development according to the GMDS, such as motor, learning, hearing, etc. Is that really lifelong influence or would the negative correlation potentially disappear? Perhaps discussion is needed.
- Sleep states (AS, QS) are actually not very well defined for neonates, where often intermediate/transitional sleep (IS) presents. It is not clear how sleep states were scored in this work, e.g. based on which standard or guidelines. Page 3 84-85, the sentence "AS and QS are precursors of adult rapid eye movement (REM) and non-REM sleep states, respectively" is not precise. The correspondence between AS/QS and REM/NREM depends on sleep scoring method. It is also the case that QS corresponds to N3 (deep) sleep for adults. The reference 10 is not supporting well the statement. This should be better described.
- For the group-by-sleep interactions, I miss the results for delta band (Fig.S2). And there are repetitions of graphs between Fig.2 and Fig.S1, and between Fig.3 and Fig.S2. Why presenting two figures? Fig.S1 and S2 can be actually shown in the main text rather than as a suppl. material.
- Fig.S2, correlation seems absent in low-frequency band (theta), and possibly delta (not shown in the paper). This should be discussed. And quiet sleep/deep sleep would probably correspond to an enhanced power of slow wave in the low frequency band, how does this relate to functional connectivity? Shorter-range for QS? And the characteristics for delta band is quite different from the other frequency bands. Please explain and discuss a bit more about these.
- Maybe I overlooked, it is not clear how the changes in f-connectivity was calculated. Fig.3 and Fig.S2, graph A and B have their vertical axis "Connectivity" and "Delta Connectivity" and for both "changes" were mentioned in the figure caption. This is a bit confusing to me. Please clarify.
- It seems that less changes in functional connectivity would have higher visual and socialemotional scores (negative correlation). Does it indicate something about stability of brain activity? Is this different between AS and QS and between term and preterm infants? I expect that AS also plays an important role for neural-development in a certain aspect, and a balance of AS and QS suppose to be optimal. Good to also present these results and corresponding discussion.
- Please provide a higher quality image of Fig. S5.
- The authors refer to a lot of tool-kits, some technical steps of using those tool-kits would be appreciated.

Reviewer #1

In this paper, the authors investigate cortical dynamics associated with sleep states in pre- and full-term infants by using a very clever connectivity analysis of EEG and more importantly a model based study of the underlying eigenmodes. The paper is, in my view, extremely relevant not only for basic neuroscience but also for clinical applications aiming to predict the development of pre-term neonates. More importantly, I think they show very specifically the necessity of using whole-brain modelling for extracting the biophysical underlying causative mechanisms of the observed changes.

In particular, they show significant changes in the global brain connectivity underlying the babies AS and QS sleep stages differentiating the control normal babies respect to the pre-term neonates. Furthermore, they were able to demonstrate that those significant changes in global connectivity underlying different sleep stages are due to a specific deficit in large-scale neuronal dynamics.

More specifically, they observed that AS sleep is associated with increased functional connectivity in visual occipital regions whereas QS is associated with a broader increase in long-range cortical connectivity.

The paper is very well written and the methodology proposed is novel and solid, so I'm absolutely convinced that this paper should be accepted in Nature Communications due to its novelty, relevance and elegant combination of empirical phenomenological analysis with causative model-based explanations.

I have only some minor remarks that the authors could decide to consider.

Authors (A): We appreciate the reviewer's positive appraisal of our work.

R1.1. I think that would be nice to analyse the data also in narrow bands (let say in windows of 2Hz in order to have a more specific frequency analysis. Even more, perhaps after the first narrow band filters, it would be nice to analyse the functional correlation of the envelopes after a second slow filter (let say from 0.01 Hz to 0.2 Hz). the motivation is the type of analysis that one use in MEG (e.g. Hipp et al. Nature Neuroscience volume 15, pages 884–890 (2012)). Perhaps one can also discover the relevant time scale (associated with this second filter), i.e. for which time scale the method is more sensitive.

A: This is an excellent suggestion. Accordingly, we have performed these analyses and now include the results in the main text as well in the revised Figure 2 and Figure 3 (previously Figure 2). Specifically:

Results section (pp5-6):

To further characterize the spectral fingerprints of these effects, we filtered the neural signals into (fast) carrier signals and their (slow) amplitude fluctuations across a staircase of relatively narrow bands (evenly spaced in logarithmic coordinates and covering the carrier-amplitude frequency space), then repeated the network-based contrasts. These analyses confirm the existence of functional network effects in both quiet and active sleep states, supported by fast carrier oscillations that are strongest in alpha and beta frequencies (**Fig. 2C**), and whose amplitudes fluctuate maximally on time scales of 5-11 seconds (0.09-0.19 Hz). The strongest effects are seen for higher carrier frequencies in QS than AS (~16 Hz vs ~10 Hz). Interestingly, there exists a second effect in QS centered over the delta band, composed of relatively short-range edges that involve frontal regions and consistent with the enhanced power of slow-wave activity classically seen in quiet/deep sleep (**Supplementary Fig. 1**).

Fig. 2. Changes in cortico-cortical functional connectivity as a function of sleep state. (A) The main effect of sleep state in the alpha band shows significant connectivity differences in two main cortical networks (red: AS>QS, blue: QS>AS; both $p_{FWER} < 0.0002$, paired two-tailed t-test). Distinct cortical regions are coded with different colors: temporal (green), central (pink), occipital (black), and frontal (orange). (B) These networks have different characteristic length distributions and distinct cortical region distributions. Curves are distributions (kernel density estimates) of functional connection lengths given by the fraction of edges p (normalized to the whole network) for AS (red) or QS (blue). Bar plots show participation of broad cortical areas in each sleep-related network. Filling of the bars denotes the percentage of involved nodes within each region and height denotes each region's share in the whole network. (C) Spectral fingerprints of AS (left) and QS (right) sleep. Color scale shows effect size (Cohen's d) of sleep contrasts centered at different carrier and amplitude frequencies (f). Darkest blue shade denotes frequency combinations where there were no suprathreshold edges (null).

Across both sleep states, EP infants show significantly stronger connectivity compared to HC with functional networks in the alpha range encompassing occipital, temporal, central, and frontal regions ($p_{FWER} < 0.0001$, for all paired two-tailed t-tests; Fig. 3A, B). These effects are also associated with stronger long-range cortico-cortical connectivity in EP relative to HC (Fig. 3B). Spectral analyses using narrow-band filters suggest this effect is strongest in the alpha and beta range (Fig. 3C), although long-range functional networks, significantly stronger in EP, are also present in the delta and theta bands (Supplementary Fig. 1). EP infants also show a reduction in connectivity between anterior and posterior cortical regions in the theta band ($p_{FWER} = 0.01$, paired two-tailed t-test; Supplementary Fig. 1). When analyzed according to the bank of narrow-band filters, the reduction in functional networks is evident

across the theta, alpha and beta frequencies (**Fig. 3C**). However, when analyzed according to the four pre-specified frequency bands, these effects only survive FWE correction in the theta range (**Supplementary Fig. 1**).

Fig. 3. Changes in cortico-cortical functional connectivity as a function of birth gestational age. **(A)** The main effect of group shows higher long-range connectivity in the alpha band in EP infants compared to HC. The corresponding widespread network is composed of temporal (green), central (pink), occipital (black), and frontal (orange) cortices ($p_{\text{FWE}} < 0.0001$, paired two-tailed t-test). **(B)** The line shows the distribution (kernel density estimate) of functional connections lengths, where p is the fraction of edges (normalized to the whole network). Bar plots show participation of broad cortical areas in each sleep-related network. **(C)** Spectral fingerprints of EP (left) and HC (right) infants. Colors show the effect size (Cohen's d) of group contrasts for all combinations of carrier vs. amplitude frequencies (f). Darkest blue shade marks the cases where there were no suprathreshold edges (null).

Methods section (pp. 16-17):

Assessment of effect size in the frequency domain

To study the spectral fingerprints of sleep and gestational age effects, pre-processed EEG data were filtered into 21 frequency bands covering the range 0.42-22 Hz. Central frequencies (f) of these filters were spaced such that the $(k + 1)$ -th frequency was a fixed multiple of the k -th frequency given by $f_{k+1} = 1.2f_k$. The slowest carrier frequency f_1 was set to 0.5 Hz and the ensuing highest frequency was $f_{21} = 19.2$ Hz. Cut-off frequencies were $0.85f$ and $1.15f$, and stop-band frequencies were $0.5f$ and $1.5f$. Band-pass filtering was implemented by applying pairs of low-pass and high-pass Butterworth filters.

This yielded carrier oscillations with overlapping frequency bands of equal width on a logarithmic scale. The amplitudes of these carrier frequencies were then band-pass filtered into 15 frequency bands to estimate amplitude fluctuations at different time scales⁴⁹. Amplitude filters were designed according to the same principles as for the carrier frequencies. The slowest amplitude frequency f_1 was set to 0.015 Hz and the ensuing highest frequency was $f_{15} = 0.19$ Hz. This double filter bank led to $21 \times 15 = 315$ functional connectivity matrices per subject at each sleep state. Connectivity analyses of the main effects of group and sleep state were estimated using NBS with a search (height) t-statistic threshold of 2.5. Frequency combinations (carrier and amplitude) for which there were no suprathreshold edges were denoted null (**Fig. 2C, 3C**). Effect size was computed as Cohen's d (i.e., NBS-derived mean t-statistic divided by the square root of the total degrees of freedom).

R1.2: I assume that based on the eigenmodes of the anatomical connectivity matrix, one could use those templates for also analyzing the correlation patterns and their changes on those modes. I mean to use the modes described in Selen et al. Nature Communications volume 7, Article number: 10340 (2016). Of course, I'm NOT asking to do these analyses, that would be a new paper, but perhaps to discuss/speculate about them in the conclusions as alternative, or to argue why functional modes are more suitable than those anatomical eigenmodes.

A: We would like to clarify that the eigenmodes we use are not purely functional but derived through an eigenmode decomposition on the cortical surface (using the Laplace-Beltrami operator; see Equation S2, Methods, pp17-18). To achieve this, we used a cortical surface scaled to the infant size and spatially smoothed to match the brain folding at term-equivalent age. Functional modes are then obtained from these eigenmodes by scaling the coefficients of these anatomical eigenmodes and multiplying the resulting pattern by the (noisy) oscillatory activity (Equation S1). Using the connectome to calculate eigenmodes, instead of the cortical surface, is an interesting proposition. Unfortunately, to the best of our knowledge, no well-validated and widely-accepted term-age neonatal connectomes currently exist. Moreover, at least in the adult, the human connectome is dominated by spatial geometry, which is why the modes of the two approaches bear close similarity. This is likely to be true – possibly even to a greater extent – in the neonate. To reflect on this in the paper, we have added the following text (p12),

Likewise, our eigenmode decomposition is currently derived from cortical geometry only: The incorporation of structural connectomes of infant brain into the decomposition¹³ could further refine and individualize the functional modes used for our biophysical model. Such work requires further advances in the accurate reconstruction of individual tractography data from neonates. Given that geometry contributes substantially to connectivity³¹, which shapes large-scale dynamics³², it is likely that such resulting modes would differ only slightly from those presently used.

Reviewer #2

The authors analyzed brain cortical connectivity between infant sleep states (active sleep/quiet sleep) and between extremely premature neonates and term infants. They also statistically analyze the sleep state-related connectivity as prognostic information of visual performance and social-emotional performance later at around 2-year age. The work was very interesting with several significant findings and the paper is well-written. Statistical analyses were appropriate.

Authors (A): We appreciate the reviewer's positive appraisal and constructive suggestions to improve the paper, all of which have been incorporated.

R2.1: It can be better explained or motivated the choice of the visual and social emotional performance at age of 2 y for analysis. There are other aspects of the mental development according to the GMDS, such as motor, learning, hearing, etc. Is that really lifelong influence or would the negative correlation potentially disappear? Perhaps discussion is needed.

A: Deficits in socio-emotional and visual outcomes are of substantial current interest as they reflect key elements of early and developing cognition, upon which later development is scaffolded. While other indices (motor/hearing/language) have been the subject of considerable past research, concern with those neurobehavioural domains has decreased with improved preterm care. Instead, there is now increasing attention to the lifelong impact of early visual abilities and social-emotional performance. To optimize the practical (clinical/translational) impact of our work, we therefore chose to assess correlations between neural activity and visual and social-emotional performance. Notably, choosing visual performance as a benchmark is also consistent with our cortical network analyses highlighting a key impact of preterm birth in occipital systems.

At this stage, it is difficult to extrapolate to lifelong effects from birth and early (2 year) outcomes, as it is likely that the observed outcomes will be subject to a number of later modifiers. Notably, the qualitative changes in cognitive abilities from early childhood to adulthood fundamentally challenge attempts to link early deviations in neurodevelopment to adult outcomes.

The methods and discussion sections have been amended to highlight this important point:

Methods (p15):

The neurodevelopment of infants was assessed at two years of corrected age using the structured Griffiths Mental Developmental Scales⁴⁰. These scales were chosen because they measure the cognitively demanding domains of visual performance and social-emotional performance. The assessment of visual and social-emotional performance in EP was chosen because of their established, widespread clinical use plus their broad impact on lifelong neurocognitive performance and quality of life, hence improving the translational potential of our study^{41,42}. While other outcomes such as gross motor development and hearing are also affected in some infants, studies in ex-preterm infants have shown that these are the most likely to be modified by a host of individual and treatment interventions⁴³.

Discussion (p12):

Future work, incorporating longer-term (life-long) outcomes and other functional metrics could also help disambiguate those preterm-related functional network deficits that persist from those that can be modulated by environmental influences during childhood (e.g. inter-personal relationships, health interventions, education).

R2.2: Sleep states (AS, QS) are actually not very well defined for neonates, where often intermediate/transitional sleep (IS) presents. It is not clear how sleep states were scored in this work, e.g. based on which standard or guidelines. Page 3 84-85, the sentence "AS and QS are precursors of adult rapid eye movement (REM) and non-REM sleep states, respectively" is not precise. The correspondence between AS/QS and REM/NREM depends on sleep scoring method. It is also the case that QS corresponds to N3 (deep) sleep for adults. The reference 10 is not supporting well the statement. This should be better described.

A: It is correct that sleep states have less standardized definitions for infants in the AASM guidelines and other clinical consensus. Moreover, the developmental trajectories of infant sleep states and links with counterparts in adults are not entirely clear. With these caveats in mind, there are also a number of convergent behavioral and neurophysiological correlates of infant sleep to support the existence of genuine sleep (active and quiet) states since the early stages of the ontogenesis^{a,b,c}. Notably, representative sleep epochs were selected from within well-established patterns of corresponding behavior and brain activity, and not close to the transition between QS and AS. We have edited the revised manuscript as follows:

The sentence referred to by the reviewer in the Introduction was edited (p3),

These distinct states of vigilance are key components of the infants' sleep-wake cycle, which gradually transform with neurodevelopment into the alternating cycle of mature rapid eye movement (REM) and non-REM sleep states such as deep sleep¹⁰.

To more clearly highlight the scoring method, we created a new subsection heading and modified the text which now reads, (Methods p14),

Sleep state selection

Each recording session continued until the infant had undergone two sleep (vigilance) states: active sleep (AS) and quiet sleep (QS). Vigilance state assessment was performed in our data through a combination of electrophysiological and behavioral measures. Polygraphic channels (chin electromyogram, electrocardiogram, electrooculogram, and respiratory sensors) were used for this purpose. EEG traces during AS exhibit continuous fluctuations, respiration is irregular, and occasional eye movements are present. Conversely, EEG during QS is characteristically discontinuous, and respiration is regular³⁹ (**Fig. 1A**). Next, we selected 5-min-long artifact-free EEG epochs from the most representative periods of AS and QS. The selection of the epoch length was based on our previous studies demonstrating that a continuous 5 minute recording duration provides a reliable and stable estimate of the whole-brain functional network activity in infants^{4,7}. To avoid transitions between vigilance states, representative sleep epochs were selected from within well-established patterns of corresponding behavior and brain activity, and not close to the transition between QS and AS.

R2.3: For the group-by-sleep interactions, I miss the results for delta band (Fig.S2). And there are repetitions of graphs between Fig.2 and Fig.S1, and between Fig.3 and Fig.S2. Why presenting two figures? Fig.S1 and S2 can be actually shown in the main text rather than as a suppl. material.

A: There is no significant interaction between sleep state and group in the delta band, which we now note (caption to Supplementary Fig. 2, p25),

There were no significant group-by-sleep interactions in the delta band.

We have also revised Supplementary Fig. 1 and S2 to remove the redundancies (i.e., the alpha band networks).

Updated Figure S1 (alpha band removed)

Updated Figure S2 (alpha band removed)

Note that, in response to suggestions by R1, we have now introduced summary “spectral fingerprints” into the main text which summarize sleep and group effects across all frequencies. As a result, we split Figure 2 into Figures 2 and 3 in the revised manuscript.

We now also note the results for the delta band (p5),

Interestingly, there exists a second effect in QS centered over the delta band, composed of relatively short-range edges that involve frontal regions and consistent with the enhanced power of slow-wave activity classically seen in quiet/deep sleep (Supplementary Fig. 1).

We considered the possibility of presenting results for all four pre-chosen frequency bands in the main text. However, with the new spectral results, we feel the effects are already well summarized in the main text. We feel that moving all four frequency band results will add redundancy and distract from the core message of the paper: We respectfully prefer to only keep the alpha-band results in the main figures, leaving the other three bands in the Supplementary Information.

R2.4: Fig.S2, correlation seems absent in low-frequency band (theta), and possibly delta (not shown in the paper). This should be discussed. And quiet sleep/deep sleep would probably correspond to an enhanced power of slow wave in the low frequency band, how does this relate to functional connectivity? Shorter-range for QS? And the characteristics for delta band is quite different from the other frequency bands. Please explain and discuss a bit more about these.

A: Yes, it is correct, as stated above, the new spectral analyses show that the QS>AS contrast shows an effect size (ES) peak centered over the delta band which we now note in the Results (p5),

Interestingly, there exists a second effect in QS centered over the delta band, composed of relatively short-range edges that involve frontal regions and consistent with the enhanced power of slow-wave activity classically seen in quiet/deep sleep (Supplementary Fig. 1).

But there is no significant interaction between sleep state and group in the delta band, which we now note (caption to Supplementary Fig. 2, p25),

There were no significant group-by-sleep interactions in the delta band.

We also highlight the null results for the low-frequency bands and other regressions in the Results (p8),

No significant associations between sleep-induced connectivity changes and visual behavior are detected for low-frequency bands. A negative correlation is also apparent for social-emotional performance, but the association is not statistically significant when accounting for multiple comparisons (all $p_{uncorrected} > 0.039$; Fig. 4B and Supplementary Fig. 2B). The regressions against socio-emotional indices and functional networks are not significant in any frequency band, nor are those for visual outcome and functional networks in the theta band (Supplementary Fig. 2B).

We also expand on these points in the Discussion (p11),

That is, alterations in functional connectivity in visual cortex due to preterm birth may impact on the maturation of functional assemblies that are important for emerging visual function. Although we find enhanced correlations in lower frequencies during QS, as might be expected for activity during deep sleep²⁴, these do not correlate significantly with neurobehavioral outcomes. Rather, our findings support the notion that synchronization within the visual system at alpha and beta frequencies is key for the development of perceptual function^{25,26}.

R2.5: Maybe I overlooked, it is not clear how the changes in f-connectivity was calculated. Fig.3 and Fig.S2, graph A and B have their vertical axis "Connectivity" and "Delta Connectivity" and for both "changes" were mentioned in the figure caption. This is a bit confusing to me. Please clarify.

A: We have edited both captions (Fig. 4, pp7-8 and Supplementary Fig. 2, p25) to make this clearer,

Correlations between changes in the mean cortical connectivity strength of the occipital pattern (connectivity strength of the orange network of panel A in AS minus QS) and neurodevelopmental outcome assessed at two years of age in EP infants.

R2.6: It seems that less changes in functional connectivity would have higher visual and social-emotional scores (negative correlation). Does it indicate something about stability of brain activity? Is this different between AS and QS and between term and preterm infants? I expect that AS also plays an important role for neural-development in a certain aspect, and a balance of AS and QS suppose to be optimal. Good to also present these results and corresponding discussion.

A: This is an interesting observation which, as suggested, we now revisit in the Discussion (p11-12),

Intriguingly, the correlation between sleep-state related functional network changes in the EP infants and visual function at 2 years (Fig. 4B) is in the opposite direction to the between-group effect with the HC infants. Without experimental control over the exposure of EP infants to normal sensory experience in the neonatal intensive care unit, it is not possible to infer whether these effects are driven by preterm birth or the additional post-natal exposures of EP infants. Future work, incorporating longer-term (life-long) outcomes and other functional metrics could also help disambiguate those preterm-related functional network deficits that persist from those that can be modulated by environmental influences during childhood (e.g. inter-personal relationships, health interventions, education).

R2.7: Please provide a higher quality image of Fig. S5.

A: As requested, we improved the quality of the image and made it more intuitive to readers by removing all redundant labels; removing two middle rows; enlarging the panels and fonts; adding labels for goodness of model fitting; adding a new legend to indicate histogram ranking; and exporting at higher quality for the Supplementary Material PDF. Please note that the quality of the figure may be reduced in the PDF generated by the submission portal.

R2.8: The authors refer to a lot of tool-kits, some technical steps of using those tool-kits would be appreciated.

A: We have now added a new Supplementary Figure S9 (see below) that shows the entire analytic pipeline, together with download links (linked to the packages names) to the original toolkits. The following sentence has been added in the "EEG data pre-processing" section (p15):

An overview of the entire analytic pipeline is shown in Supplementary Fig. 9.

Note that, as per the journal requirements, we have provided all in-house customized MATLAB code for running our analyses (shared via GitHub). We also added a sentence to the new 'Code availability' section (p19):

Custom Matlab code implementing connectivity analysis, the biophysical model, and its fitting to empirical data can be found here: <https://github.com/babyEEG/Infant-Sleep>.

Supplementary Fig. 9. Overview of the analytical pipeline. This schematic shows the major processing stages (orange boxes) of the data processing. Software packages that were used at each stage are indicated in the green boxes (with hyperlinks to the corresponding download sites).

References

- Karlsson, K.A., Gall, A.J., Mohns, E.J., Seelke, A.M. & Blumberg, M.S. The neural substrates of infant sleep in rats. *PLoS Biol* **3**, e143 (2005).
- Seelke, A.M. & Blumberg, M.S. The microstructure of active and quiet sleep as cortical delta activity emerges in infant rats. *Sleep* **31**, 691-699 (2008).
- Seelke, A.M., Karlsson, K.A., Gall, A.J. & Blumberg, M.S. Extraocular muscle activity, rapid eye movements and the development of active and quiet sleep. *Eur J Neurosci* **22**, 911-920 (2005).

REVIEWERS' COMMENTS:

Reviewer #1 (Remarks to the Author):

The authors considered all the comments, excellent job! The present version is improved and ready for publication.

Reviewer #2 (Remarks to the Author):

My comments have been well addressed, thanks.